# A general strategy for the ultrafast surface modification of metals

Mingli Shen[1,*], Shenglong Zhu[1,*] & Fuhui Wang[1,*]

Surface modification is an essential step in engineering materials that can withstand the increasingly aggressive environments encountered in various modern energy-conversion systems and chemical processing industries. However, most traditional technologies exhibit disadvantages such as slow diffusion kinetics, processing difficulties or compatibility issues. Here, we present a general strategy for the ultrafast surface modification of metals inspired by electromigration, using aluminizing austenitic stainless steel as an example. Our strategy facilitates the rapid formation of a favourable ductile surface layer composed of FeCrAl or β-FeAl within only 10 min compared with several hours in conventional processes. This result indicates that electromigration can be used to achieve the ultrafast surface modification of metals and can overcome the limitations of traditional technologies. This strategy could be used to aluminize ultra-supercritical steam tubing to withstand aggressive oxidizing environments.

[1] Laboratory for Corrosion and Protection of Metals, Institute of Metal Research, Chinese Academy of Sciences, 62 Wencui Road, 110016 Shenyang, China. * These authors contributed equally to this work. Correspondence and requests for materials should be addressed to M.S. (email: mlshen@imr.ac.cn).

The development of new surface modification technologies is essential to fabricate materials that can withstand the increasingly aggressive environments encountered in various modern energy-conversion systems and chemical processing industries. For example, aluminizing, which can substantially improve the high-temperature oxidation resistance of substrate metals at low cost, could in principle cost-effectively enhance the reliability and lifetime of the heat-resistant steels used for high-temperature/pressure steam tubing in supercritical (SC) or ultra-SC water-based fossil-fired or nuclear power plants[1–5]. However, most traditional and newly developed technologies are frequently impractical to use due to limitations such as slow diffusion kinetics, processing difficulties or compatibility issues[1,4,6–8].

Here we present an easy-to-operate, general strategy for ultrafast surface modification of metals inspired by electromigration[9–14]. We use this strategy to aluminize an austenitic stainless steel (304SS)—a prototype material for high-temperature/pressure steam tubing—as an example. The extremely fast electromigration-assisted aluminizing (EMAA) was conducted by passing a pulsed electric current through the steel. We find that this regime facilitates the outward diffusion of Fe in the steel, resulting in the rapid formation of a favourable ductile surface layer composed of FeCrAl or β-FeAl within only 10 min by EMAA, whereas forming this layer requires several hours in conventional processes. Our results indicate that electromigration can be used to achieve the ultrafast surface modification of metals and to overcome the disadvantages of traditional technologies.

## Results

**Formation of aluminized layers.** We demonstrate the dramatic benefits of incorporating electromigration into surface modification processes, using the aluminizing of an austenitic stainless steel (304SS) as an example. The aluminizing process consists of passing electric currents through the samples (1 mm × 10 mm × 100 mm in size), and the proposed current flow patterns are illustrated in Fig. 1. The driving force for aluminizing, that is, the chemical potential gradient (CPG) is conventionally known to occur in a direction perpendicular to the surface of the substrate metals. In direct current (DC) mode, the resultant electromigration force (EMF) flows parallel to the metal surface and is thus unable to couple with the CPG, whereas in pulsed DC (PDC) or alternating current (AC) mode, the changing current produces a changing magnetic field, and an eddy current is thus induced in the metal according to Faraday's law. The self-induced eddy current is largest near the metal surface where the aluminizing takes place. In principle, a heterogeneous medium is more favourable for an eddy current to flow perpendicularly to the metal surface[15]. Aluminizing, which produces a compositional gradient 'skin' on the metal, may create this favourable heterogeneity. Thus, the eddy current that flows in the heterogeneous 'skin' of metals enables coupling between the EMF and CPG[15], which would offer an alternative way to alter the aluminizing process in addition to temperature.

To verify this proposal, we aluminized a sample by passing DC (DC-aluminizing) and PDC (PDC-aluminizing) or AC currents through Al slurry-covered 304SS samples at room temperature. A DC current of ~1,300 A cm$^{-2}$ and a PDC current of ~1,400 A cm$^{-2}$ both increased the temperature to ~860 °C, as measured by type-K thermocouples on the sample. On applying the DC current for 10 min, we obtained several dark grey surface layers, which were ~35 μm thick in total (Fig. 2a). The layers are composed of an outermost hard layer of $Fe_4Al_{13}$, an adjacent hard layer of η-$Fe_2Al_5$ with dispersed $Cr_xAl_y$ particles[16], and two thin

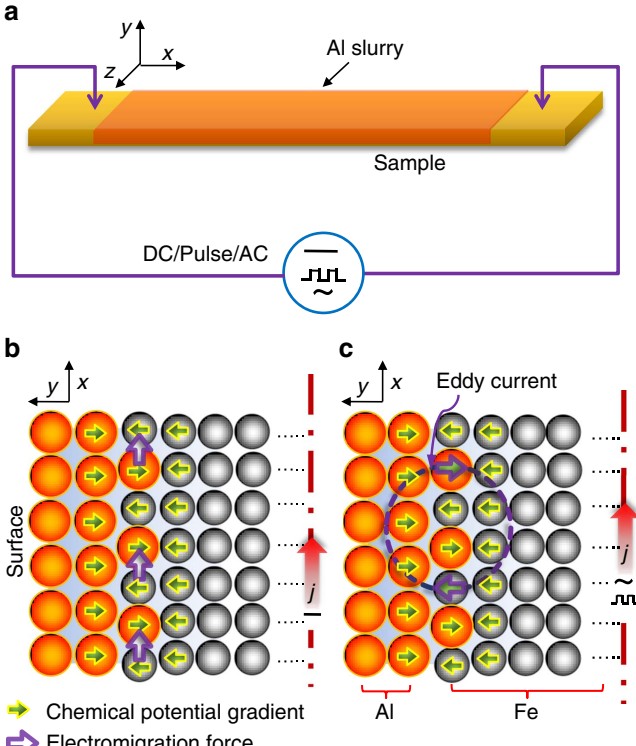

**Figure 1 | EMAA.** (**a**) Schematic illustration of the set-up for aluminizing by passing electric currents through the sample. Proposed current flow patterns: (**b**) DC and (**c**) PDC or AC.

softer inner layers of β-FeAl and FeCrAl (more specifically, a solid solution of Fe–Cr–Al and a small amount of Ni) as determined by a combination of energy dispersive spectroscopy (EDS, Fe in the balance), X-ray diffraction (XRD) analysis and indentation tests. Such a layered structure is a disappointing result and is due to the predominant inward-diffusion of Al, which also occurs when traditional methods are performed at 860–900 °C either for 10 min (Supplementary Fig. 1) or for 5 h[6]. The growth kinetics of the layers produced by DC-aluminizing are identical to their counterparts produced by using traditional methods, indicating that the DC current only affect the aluminizing by Joule heating. By contrast, for the PDC-aluminized (10 min) samples, we also observed ~35 μm thick layers which exhibit a dramatically different layer structure which is composed of only two softer layers, an outer β-FeAl and an inner FeCrAl layer doped with tiny nitride precipitates, as determined by the same analytical methods mentioned above (Fig. 2b). Evidently, the β-FeAl and FeCrAl phases preferentially grew during this process, revealing that the outward diffusion of Fe predominated with this aluminizing mode[17].

**Growth kinetics.** To assess the layers' growth kinetics[18], we measured the thickness of the layers over time in both the DC- and PDC-aluminized samples. We obtained a similar parabolic growth pattern but with substantially different kinetics. For example, a nominal interdiffusion coefficient of ~5.2 × 10$^{-15}$ m$^2$ s$^{-1}$ is obtained for the β-FeAl layer in the DC- or traditionally aluminized samples[17], whereas the value for the layer in the PDC-aluminized samples is ~5.5 × 10$^{-13}$ m$^2$ s$^{-1}$, more than two orders of magnitude higher than that for the DC- or traditionally aluminized samples (Supplementary Fig. 2). Similar results were also observed on the FeCrAl layer (Fig. 2d).

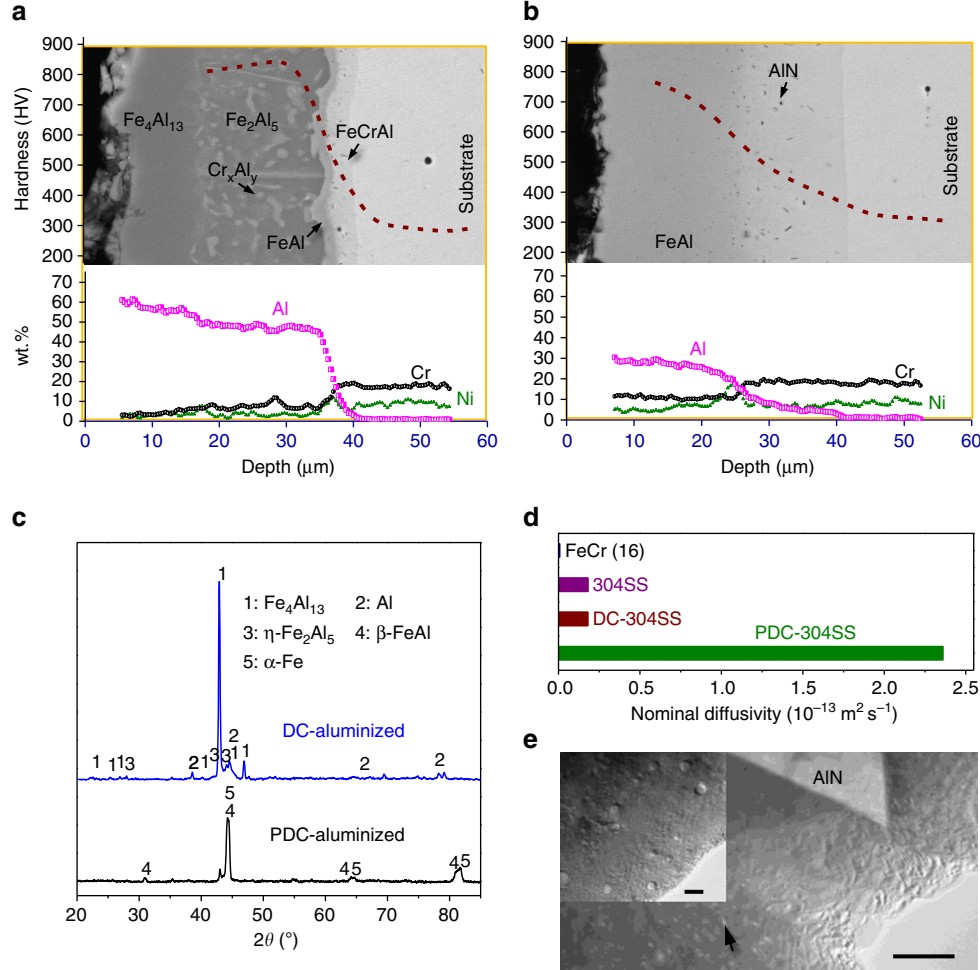

**Figure 2 | Characterization of the aluminized layer.** SEM image, EDS composition and indentation depth profile of the cross-section of the aluminized layer after applying (**a**) DC at 1,300 A cm$^{-2}$ for 10 min, (**b**) PDC at 1,400 A cm$^{-2}$ for 10 min. (**c**) XRD patterns of the two samples (**a,b**). (**d**) Nominal diffusivity in the FeCrAl phase, showing a significantly higher growth rate of the FeCrAl layer in the PDC aluminized sample. (**e**) TEM image of the inner layer of the PDC aluminized sample (**b**). Scale bars in **e**, 200 nm; 20 nm (inset).

Moreover, substantial nanoscale voids or vacancy clusters dispersed in the FeCrAl layer of the PDC-aluminized samples were observed by transmission electron microscopy (TEM) bright-field imaging (Fig. 2e), indicating that the outward diffusion of Fe is predominant in the process because in diffusion couples, voids can only be segregated in the branch with faster diffusing atoms to balance the deficiency of slowly diffusing atoms, according to the Kirkendall effect[19]. The voids are distributed randomly in a localized area of the layer, and are mostly <20 nm in size. Such a distribution might be attributed to the concurrent accumulation and annihilation of the voids at high temperatures. Similar β-FeAl and FeCrAl layers were also obtained when an AC current (1,200 A cm$^{-2}$, 10 min) was applied (Supplementary Fig. 3), which provides further evidence of the necessity of an alterant current in facilitating coupling between the EMF and the CPG. No abnormal grain growth or microstructure deterioration was observed in the substrate after aluminizing by passing the alterant current[20] (Supplementary Fig. 4).

**Rapid growth of FeCrAl monolayer.** We further applied a higher PDC of ∼1,900 A cm$^{-2}$ for only 5 min and again obtained a ∼35 µm thick aluminized layer, but it was only composed of a soft FeCrAl monolayer of the α-Fe phase, as determined by selected-area electron diffraction (SAED) and the other analytical

methods mentioned above (Fig. 3). The higher PDC showed a stronger coupling due to the higher current applied. Despite the greater temperature rise (to ∼1,050 °C) on the substrate, the extremely high growth rate of this layer on the entire surface of the sample has not been achieved by any other aluminizing methods thus far. For example, hot-dipping, which is the fastest method for aluminizing large-scale components, requires ∼60 min of annealing at 1,050 °C to obtain a similar solid solution layer[5].

**Oxidation behaviour.** To assess the Al$_2$O$_3$-forming abilities, we exposed sectioned samples that were cut from the PDC-aluminized 304SS plates with a bilayer of β-FeAl/FeCrAl (1,400 A cm$^{-2}$, 10 min) or with a monolayer of FeCrAl (1,900 A cm$^{-2}$, 5 min) at 700 °C and 900 °C in oxygen with 40–50 vol % (volume per cent) water vapour, an environment that is extremely aggressive to conventional Cr$_2$O$_3$-forming alloys (Fig. 4). Thermogravimetric analysis shows very high specific-mass gains for the naked 304 substrate. The Cr oxy-hydroxide species vaporized at the initial exposure stage as indicated by the mass losses of 304SS after 20 min at 900 °C and after ∼25 h at 700 °C, followed by catastrophic oxidation, as indicated by the huge specific-mass gains subsequently observed[1,21]. By contrast, the aluminized samples showed substantially lower specific-mass gains due to the formation of a protective Al$_2$O$_3$ scale.

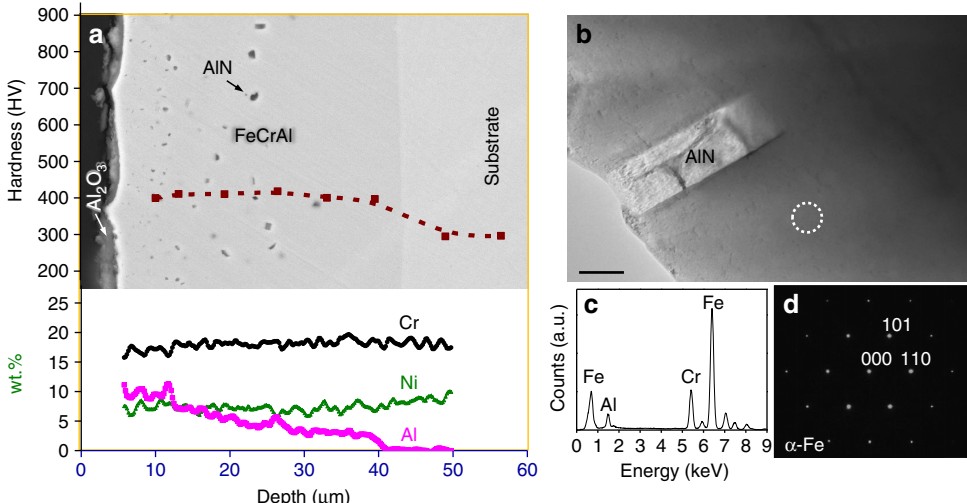

**Figure 3 | Ultrafast formation of the FeCrAl layer.** (**a**) SEM image, EDS composition and indentation depth profile of the cross-section of the aluminized layer after applying PDC at 1,900 A cm$^{-2}$ for 5 min. (**b**) TEM image of the FeCrAl layer as shown in **a**; scale bar: 100 nm. (**c**) EDS pattern, and (**d**) Corresponding SAED in the region denoted by a dashed circle in **b**.

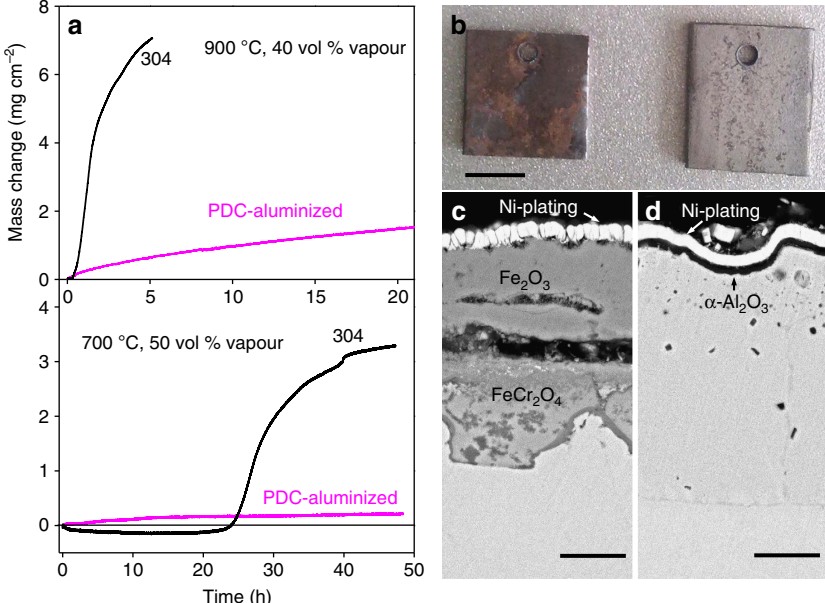

**Figure 4 | High-temperature oxidation in water vapour.** (**a**) Oxidation kinetic curves of pristine and aluminized 304SS samples: the bilayer sample of β-FeAl/FeCrAl obtained with PDC at 1,400 A cm$^{-2}$ for 10 min was tested in oxygen with 40 vol % (volume per cent) water vapour at 900 °C, and the monolayer sample of FeCrAl obtained with PDC at 1,900 A cm$^{-2}$ for 5 min was tested in oxygen with 50 vol % water vapour at 700 °C. (**b**) Photographs of the 304SS (left) and aluminized (right) samples after the test at 900 °C. SEM images of cross-sections of oxide scales formed on (**c**) the 304SS, and (**d**) the aluminized samples after the test at 700 °C. Scale bars: (**b**) 5 mm; (**c**, **d**) 10 μm.

## Discussion

The opportunity offered by eddy currents to speed up and tailor the aluminizing process through coupling between the CPG and EMF may also be described by the atom diffusion flux $J_a$ (refs 9–11)

$$J_a = J_{chem} + J_{em} = D_a\left(-\frac{\partial C_a}{\partial x}\right) + C_a \frac{D_a}{kT}Z^* ej\rho \qquad (1)$$

where $J_{chem}$ and $J_{em}$ are the atomic fluxes driven by the concentration gradient (or more precisely the CPG) and EMF, respectively; $C_a$ and $D_a$ are the concentration and diffusivity, respectively; $Z^*$ is the effective charge number; $e$ is the electron charge; $j$ is the current density and $\rho$ is the resistivity. In DC mode, there is no current that flows parallel to the CPG, that is

$j = 0$, so the atom fluxes for Al and Fe during the DC-aluminizing are reasonably determined by the CPG. By contrast, in PDC mode, $j$ is no longer zero at the time instant of each rising and falling of the current pulse. The EMF would couple with the CPG in a discontinuous manner to co-drive the atomic migration. The exact magnitude for $j$ remains unknown, although it may be estimated by solving the Maxwell equations in anisotropic solids[15], but we may still qualitatively evaluate the influences of $j$ on the atom fluxes of Al and Fe. In simple terms, there is a proportional relation between $j$ and the time dependence of the applied current d$j_{ap}$/d$t$. According to Ampère's law, the applied time-dependent current d$j_{ap}$/d$t$ produces a time-dependent magnetic field d$B$/d$t$ that is numerically proportional to d$j_{ap}$/d$t$. Meanwhile, the changing magnetic field generates an eddy

current $j_{ed}$ that is numerically proportional to $dB/dt$, according to Faraday's law. This result occurs because the current $j$ is one branch of the eddy current $j_{ed}$ that flows perpendicularly to the metal surface. A simplified relation can be obtained wherein $j$ is proportional to the time dependence of the applied current $dj_{ap}/dt$. The degree of the influence is closely related to the exact $Z^{\star}$ value for a specific element. A calculation of the $Z^{\star}$ values for Al ($-11.2$) and Fe ($-257.5$) showed a substantial difference between the two constituents[22], revealing that the current could induce a much greater atom flux of Fe than Al (that is, the dramatically enhanced outward diffusion of Fe would occur), which is consistent with the observation of vacancy clusters present in the PDC aluminized layers. Moreover, most of the electromigration-induced mass transport shows a polarity effect, which has been observed in a considerable number of experiments wherein a pre-existing interfacial intermetallic compound or solid solution layer is hardly decomposed as the current reverses its direction[23–25]. We observed an increasingly faster growing aluminide layer induced by the eddy current that frequently alters its direction between the rise and fall of a single pulse, which also demonstrates this feature[26]. Although the observed diffusion behaviour can be qualitatively interpreted by the coupling effect, other possible effects cannot be ruled out. For example, a phonon effect, in which lattice vibrations are triggered by an applied alternative electromagnetic field, has been demonstrated to favour atomic diffusion by lowering the migration barriers in the intermetallic compound[27]. Both a lower migration barrier and higher vacancy concentration contribute to the faster atom diffusion. As a diffusion driving force, phonons have only been reported to favour atomic diffusion in intermetallic phases, which is less likely to explain the enhanced diffusion observed in solid solution phase such as FeCrAl. In addition, thermomigration induced by temperature gradient in the sample was not considered because only a negligible temperature gradient can be established in the sample[13].

The strategy of coupling the CPG and EMF to achieve ultrafast surface modification, as demonstrated by aluminizing 304SS austenite stainless steels, can preferentially obtain ductile phases and substantially shorten the processing time from hours to minutes, which would improve the aluminizing of ultra-SC steam tubings in energy-conversion systems and other components in the chemical processing industries. It has also been verified in the siliconizing and chromizing of mild steels. Clearly, this strategy is a marked advance in surface modification technology.

## Methods

**Materials.** Commercial 304SS austenite stainless steel plate substrates (1 mm × 10 mm × 100 mm in size) were brushed with an Al slurry for aluminizing. The chemical composition is C 0.07, N 0.1, Mn 2.00, P 0.045, S 0.030, Si 0.075, Cr 17.5–19.5 and Ni 8.0–10.5 in weight per cent, as reported by the manufacturer (Baosteel, Shanghai, China). Before brushing with the slurry, all of the substrates were sand-blasted and ultrasonically rinsed in acetone to obtain clean surfaces. Then an Al slurry composed of pure Al powder (10 μm in mean size, with an oxygen content <0.062 wt %) and an aqueous adhesive was brushed onto all surfaces of the substrate to a thickness of ∼100 μm, followed by curing for 30 min from 60 °C to 150 °C in air.

**Aluminizing process.** PDC or AC was passed through the samples to generate a self-induced eddy current and to drive the aluminizing process. The frequencies of the pulse and AC current were 26 kHz and 68 kHz, respectively, and they induced a calculated skin depth ranging from 1.6 mm to 3.6 mm in the 304SS stainless steel. Therefore, for the 1 mm thick sample used in this study, the current of the selected frequencies and the induced Joule heat were approximately uniformly distributed throughout the cross-section of the sample. Current densities of ∼1,400 A cm$^{-2}$ and 1,200 A cm$^{-2}$ for the PDC and AC, respectively, produced a similar surface temperature of 860 ± 10 °C on the samples, as measured by a type-K thermocouple (0.15 mm in diameter) welded onto their surfaces. A DC of ∼1,300 A cm$^{-2}$ that

produced the same temperature was also used for the aluminizing processing. The voltage across the sample was ∼2 V. All the processing was carried out for 5 to 20 min in air without inert gas protection. For comparison, a traditional aluminizing processing using a furnace to heat the samples was also carried out at the same temperature and over the same period. After aluminizing, parts were cut from near the centre (10 mm in length) for testing and analysis.

**Microstructure characterization.** XRD (X' Pert PRO, PANalytical Co., Almelo, the Netherlands, Cu Kα radiation at 40 kV) was used to identify the phases in the aluminized samples before and after the oxidation tests were performed. Scanning electron microscopy (SEM; InspectF 50, FEI Co., Hillsboro, OR) and EDS (INCA, X-Max, Oxford instruments Co., Oxford, U.K.) were used to analyse the surface morphology, cross-sectional profile and composition of the aluminized layers. A TEM (JEOL JEM 2010F) equipped with EDS ('Tracor' EDX spectrometer) and SAED was used to determine the microstructure of the aluminized layers on the nanometre scale. The metallographic examination of the grain sizes of the 304SS substrates after the aluminizing was carried out by optical microscopy (Zeiss Axio observer Z1m). Before the examination, electrolytic oxalic acid etching was conducted for 20 s.

**Microhardness and high-temperature oxidation tests.** The microhardness depth profiles of the aluminized layers were obtained using a Vickers hardness tester (Buehler Micromet 5114) with a 10 g load for 10 s. The high-temperature water vapour resistance of the samples was evaluated at 700 °C and 900 °C in pure oxygen with 40–50 vol % water vapours (volume per cent) by a thermogravimetric analyzer (Sartorius 4410).

**Data Availability.** The authors declare that the data supporting the findings of this study are available from the corresponding author.

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

## Acknowledgements

We acknowledge the financial support of the National Key Basic Research Program ('973' Program) of China (grants 2012CB625100 and 2010CB631206), the National High Technology Research and Development Program ('863'Program) of China (grant 2012AA03A512) and the National Natural Science Foundation of China (grant 51301185). We thank Professor Weitao Wu for valuable discussions.

## Author contributions

M.S. conceived the idea and initiated the work. All three authors discussed the results. M.S. and S.Z. drafted the manuscript.

## Additional information

**Competing financial interests**: The authors declare no competing financial interests.

**How to cite this article**: Shen, M. *et al.* A general strategy for the ultrafast surface modification of metals. *Nat. Commun.* **7,** 13797 doi: 10.1038/ncomms13797 (2016).

**Publisher's note**: 

