## [Peer Review File · Nature Communications]

PEER REVIEW FILE

Reviewers' comments:

Reviewer #1 (Remarks to the Author):

I have reviewed the revised version of this paper and the author's responses to the earlier review for all 3 reviewer sets. The authors have made a good faith attempt to consider and address the comments brought up in the initial review process. This is very interesting and well done work. I believe it meets the criteria set forth in the Nature Communications guidelines. My recommendation is to accept for publication.

Reviewer #2 (Remarks to the Author):

This manuscript reports an ultrafast surface modification of metals by electromigration. The authors applied a pulsed DC of current density about 1400 A/cm² at room temperature for about 10 min in an austenitic stainless steel of size 1mm x 10mm x 100mm, covered by Al slurry. They found phase changes in forming FeAl and FeCrAl surface layers up to 35 μm in thickness. After that, the sample was oxidized at 700 to 900 {degree sign}C in oxygen with 40-50 % water to form a protective Al₂O₃ scale. In comparison, the conventional method by heat-treatment will take several hours to do so. The authors suggest that they achieved an ultrafast method of surface modification for applications to high-temperature/pressure steam tubing, for example. To explain the ultrafast process, the authors suggest, in Eq. (1) on p.11, a coupling between chemical potential gradient force and electromigration force under eddy current near the surface.

While the finding seems interesting, the manuscript is rejected for the following reasons.

(1) Consider the applied current (not current density), in a test sample of cross-sectional area of 1 mm x 10 mm and the given current density of 1400 A/cm², it is 140 A. Then, if we apply this current density to a steam tubing with a linear dimension of 10 times larger than the test

sample, the applied current will be about 14,000 A, which is too high to be of practical use. Rarely any company will try it.

(2) In Eq. (1), the chemical potential gradient force is a vector along the normal of the sample surface, yet the electromigration force due to eddy current has a random direction as shown in Fig. 1. How can they couple is unclear. Whether the authors can calculate the current density of eddy current from Ref. 15 is also unclear.

(3) The surface temperature was about 860 {degree sign}C during their testing. The temperature is above the melting point of Al in the Al-slurry. Electromigration may occur in molten Al under the current density of 1400 A/cm². But it may not occur in FeAl and FeCrAl layer, which most likely is in solid state near 860 {degree sign}C. The authors need to verify that electromigration can occur in these alloys at the high temperature and the low current density.

(4) The sample geometry is asymmetrical because the Al-slurry was applied on the top surface only. Thus, Joule heating and heat dissipation in the sample is asymmetrical, so there could be a temperature gradient in the sample. The authors have not considered thermomigration as a driving force in the chemical reactions in the near surface layers.

Reviewer #3 (Remarks to the Author):

My concern with the manuscript surrounds Fig. 1 and Eq. 1 on pg. 11 and their relevance to the surface modification process introduced in the manuscript.

The claim in the manuscript on pg. 3 is that "In a DC mode, the resultant electromigration force (EMF) flows in parallel with metal surface, thus having little chance to couple with the CPG; whereas in a pulsed DC (PDC) or alternating current (AC) mode, the self-induced eddy current that flows in the heterogeneous "skin" of metals has greater chances to make coupling between the EMF and CPG[15],..." the authors then introduce Fig 1 to show how these eddy currents flow. It is not clear to me why an eddy current flows between the slurry and the sample for an AC signal and not a DC nor have the authors adequately explained this phenomenon. Terms like "little chance to couple" and "greater chances to make coupling" are vague unscientific terms. They cite reference [15] which is an obscure book that I do not have access too. For an eddy current to flow there would need to be local potential differences between the slurry and sample and the authors do not explain why or how this occurs for AC and not DC. One might guess that there is a capacitive charging and discharging in the slurry (locally) that lags the sample but no rationale is given and the authors must provide one.

As it relates to Eq. 1. again, the authors cite ref. [15]. However, looking at this equation it appears to be derived from Fick's first law of diffusion. Further, as the difference between AC

and DC is the time dependence of the potential one might propose that there is also a concentration dependence with time and Fick's second law would govern ($\frac{dc}{dt} = -D \frac{d^2C}{dx^2}$). Further still, I see no time dependent potential in this relationship. The authors introduce the parameter j as:

"...the current density.... In the DC mode, there is no couplable j , so the atom fluxes for Al and Fe during DC-aluminizing are reasonably determined by the CPG. Whereas in the PDC mode, j is no longer zero. Although an exact magnitude for j is still missing, which may be estimated by solving the Maxwell equations in anisotropic solids[15], we may still gain a qualitative evaluation about the influences of j on the atom fluxes of Al and Fe."

Again, ref [15] is cited and no rationale is given for why this parameter j is a function of dE/dt (AC vs DC). Instead, vague terms are used to explain this parameter ("no couplable j ").

Manuscript: “A general strategy for ultrafast surface modification of metals” by

Mingli Shen, Shenglong Zhu, Fuhui Wang

Dear Editor

Thank you very much for your letter and your useful suggestions. We have revised our manuscript carefully according to the reviewer comments and suggestions. The main revisions are highlighted by orange colour in the revised manuscript and our responses to the comments are shown below point by point:

Reviewers' comments:

Reviewer #1 (Remarks to the Author):

I have reviewed the revised version of this paper and the author's responses to the earlier review for all 3 reviewer sets. The authors have made a good faith attempt to consider and address the comments brought up in the initial review process. This is very interesting and well done work. I believe it meets the criteria set forth in the Nature Communications guidelines. My recommendation is to accept for publication.

Reviewer #2 (Remarks to the Author):

This manuscript reports an ultrafast surface modification of metals by electromigration. The authors applied a pulsed DC of current density about 1400 A/cm² at room temperature for about 10 min in an austenitic stainless steel of size 1mm x 10mm x 100mm, covered by Al slurry. They found phase changes in forming FeAl and FeCrAl surface layers up to 35 μm in thickness. After that, the sample was oxidized at 700 to 900 {degree sign}C in oxygen with 40-50 % water to form a protective Al₂O₃ scale. In comparison, the conventional method by heat-treatment will take several hours to do so. The authors suggest that they achieved an ultrafast method of surface modification for applications to high-temperature/pressure steam tubing, for example. To explain the ultrafast process, the authors suggest, in Eq. (1) on p.11, a coupling between chemical potential gradient force and electromigration force under eddy current near the surface. While the finding seems interesting, the manuscript is rejected for the following reasons.

(1) Consider the applied current (not current density), in a test sample of cross-sectional area of 1 mm x 10 mm and the given current density of 1400 A/cm², it is 140 A. Then, if we apply this current density to a steam tubing with a linear dimension of 10 times larger than the test sample, the applied current will be about 14,000 A, which is too high to be of practical use. Rarely any company will try it.

The current required for steam tubing with a linear dimension of 10 times larger than the test sample is around 10 kA. This magnitude of current is acceptable in industry, because the current output of many commercial power suppliers used in materials manufacturing industry is in the range of 10-100 kA, such as spark plasma sintering (SPS, <http://www.thermaltechnology.com/spark-plasma-sintering.html>), and electroslag remelting (ESR, http://sxkydl.cn/ProductShow_en.asp?ID=112&TypeID=38). Moreover, it worth to mention that the voltage across the test sample of 1 mm x 10 mm x 100 mm with the passage of the current density of 1400 A/cm² is only around 2 V. If the cross-sectional area is enlarged by 100 times, the voltage would be still around 2 V for the tube with a length of 10 m. In comparison, the voltage required for ESR, for instance, would be 40-100 V under the same level of current output. This means that less power is needed for such surface modification process than that in those materials manufacturing process. On account of the very limited processing time, it is also cost effective as compared with those of conventional surface modification methods. Hence, this technique is safe and cost-effective and would be acceptable for practical use.

(2) In Eq. (1), the chemical potential gradient force is a vector along the normal of the sample surface, yet the electromigration force due to eddy current has a random direction as shown in Fig. 1. How can they couple is unclear. Whether the authors can calculate the current density of eddy current from Ref. 15 is also unclear.

The eddy current-induced electromigration force does change directions with time due to repetitive rising and falling of the current pulse. It is considered that the coupling state can only be maintained instantly during each pulse rising or falling. In this way, the eddy current-induced electromigration force would discontinuously

couple the chemical potential gradient force to co-drive the atomic migration. The coupling state is not formed at any time instant. Ref. 15 gives a formal route for the calculation. But it is hard to obtain a quantitatively accurate value for the eddy current in this case, due to lack of exact values of related physical parameters of the materials with changing compositions at high temperatures. A simplified relation between the eddy current and the applied current can be drawn according to Ampère's law and Faraday's law. The magnitude of eddy current is proportional to the changing rate of magnetic field that is generated by the alternating current (PDC or AC). On account of the high frequency of the applied PDC or AC current, the value could be roughly comparable to that of the applied current density.

(3) The surface temperature was about 860 °C during their testing. The temperature is above the melting point of Al in the Al-slurry. Electromigration may occur in molten Al under the current density of 1400 A/cm². But it may not occur in FeAl and FeCrAl layer, which most likely is in solid state near 860 °C. The authors need to verify that electromigration can occur in these alloys at the high temperature and the low current density.

The overall thickness of the Al-slurry is only about 100 μm. The melted Al film would stand only very short time at high temperatures. There are two reasons for this. One is that liquid Al infiltrate extremely fast into steel. The other is that concurrent oxidation of Al in the slurry occurs during aluminizing due to that the Al-slurry is directly exposed to the air. Hence, in most of the processing time, the electric current flows in the substrate metal other than the melted Al film. In addition, there are some solid filler in the slurry, which would substantially increase the electric resistance of the thin Al-slurry. Hence, even in the time period of presence of melted Al film, the current would be distributed much more in the substrate metal. As compared with the thin FeAl layer formed on the sample by passage of DC current, the PDC-aluminized layer of thick FeAl is considered to be resulted from electromigration. The occurrence of electromigration in solid FeAl phase has also been indicated in preferential formation of bulk FeAl intermetallics at high temperatures by

passage of a DC current of 31.8 A/cm^2 during sintering of Fe and Al powders as reported in Ref. 14.

(4) The sample geometry is asymmetrical because the Al-slurry was applied on the top surface only. Thus, Joule heating and heat dissipation in the sample is asymmetrical, so there could be a temperature gradient in the sample. The authors have not considered thermomigration as a driving force in the chemical reactions in the near surface layers.

Sorry for the less detailed description of the process. The Al-slurry was actually brushed on all sides of the substrate metal, which has been clarified in the experimental section now. In principle, temperature gradient can be caused by difference in ohm resistance of different phases in the sample during passage of current. However, the magnitude of the temperature gradient can be very low due to the high thermal conductivity of metals. For instance, according to the calculation on the temperature gradient generated by different ohm resistance in Al/Ni systems (Ref. 13), the value is only $0.07 \text{ }^\circ\text{C/mm}$. Hence, thermomigration effect was not considered as a driving force in the chemical reactions in the near surface layers.

Reviewer #3 (Remarks to the Author):

My concern with the manuscript surrounds Fig. 1 and Eq. 1 on pg. 11 and their relevance to the surface modification process introduced in the manuscript.

The claim in the manuscript on pg. 3 is that "In a DC mode, the resultant electromigration force (EMF) flows in parallel with metal surface, thus having little chance to couple with the CPG; whereas in a pulsed DC (PDC) or alternating current (AC) mode, the self-induced eddy current that flows in the heterogeneous "skin" of metals has greater chances to make coupling between the EMF and CPG[15],..." the authors then introduce Fig 1 to show how these eddy currents flow. It is not clear to me why an eddy current flows between the slurry and the sample for an AC signal and not a DC nor have the authors adequately explained this phenomenon.

Further explanation has been added in the manuscript now. It is considered that the eddy current generated by an alterant current (PDC or AC) flows mainly in

the “skin” of the metal where aluminizing occurs during the processing. In an alternating mode, the changing current produces a changing magnetic field, and in turn, eddy current is thus induced by the changing magnetic field in the metal according to the Faraday’s law. The self-induced eddy current is largest near the metal surface where aluminizing takes place. In principle, a heterogeneous medium is more favorable for eddy current to flow perpendicularly to the metal surface. Aluminizing which produces a compositional gradient “skin” on the metal may fulfill this situation. Thus, the eddy current that flows in the heterogeneous “skin” of metals is able to make coupling between the EMF and CPG. In contrast, it is known that a constant DC current produces a constant magnetic field that no eddy current can thus be generated. In addition, the DC current flows in a direction parallel to the metal surface, whereas the aluminizing occurs perpendicularly to the metal surface. Hence, the chemical potential gradient force for aluminizing can not be coupled by electromigration force in the DC mode.

Terms like "little chance to couple" and "greater chances to make coupling" are vague unscientific terms.

These terms have been modified in the manuscript now.

They cite reference [15] which is an obscure book that I do not have access to. For an eddy current to flow there would need to be local potential differences between the slurry and sample and the authors do not explain why or how this occurs for AC and not DC. One might guess that there is a capacitive charging and discharging in the slurry (locally) that lags the sample but no rationale is given and the authors must provide one.

The reference [15] describes the situation of eddy current to flow perpendicularly to a conductor surface. The book can be found in Amazon (https://www.amazon.cn/Physical-Kinetics-Volume-10-Pitaevskii-L-P/dp/0750626356/ref=sr_1_25?ie=UTF8&qid=1471406859&sr=8-25&keywords=Physical+Kinetics). Physically, the electric field produced by a changing magnetic field is a nonconservative field which is completely different from

a normal electric field produced by electric charges at rest. A potential drop can only be defined in the later electric charges produced conservative electric field. It is unable to define a potential drop for the eddy current (Giancoli, D. C. Physics for scientists and engineers with modern physics, 3rd edition. Pearson Education, 2005). Hence, there should be no capacitive charging and discharging in the slurry that contributes to the aluminizing process.

As it relates to Eq. 1. again, the authors cite ref. [15]. However, looking at this equation it appears to be derived from Fick's first law of diffusion.

The first item of Eq. 1 is based on the Fick's first law of diffusion. The second item of Eq. 1 is based on electromigration. The ref. [15] is introduced to indicate the situation how the component of eddy current flows perpendicular to a conductor surface.

Further, as the difference between AC and DC is the time dependence of the potential one might propose that there is also a concentration dependence with time and Fick's second law would govern ($dc/dt = -dJ/dx = D \cdot d^2C/dx^2$).

The potential of the applied current changes several tens of thousands of times per seconds (26-68 kHz) due to the fast moving free electrons in the metal, whereas the chemical concentration change that is driven by less moving lattice atoms would be in much lower rate. We believe that a perfect description of the overall diffusion process should follow the Fick's second law. In the mean time, a comprehensive description of the overall process should consider the phase transformation involved. However, in the short processing time, it might be reasonable to consider the process as steady state which can be appropriately simplified by use of the Fick's first law. This is also favored to understand the physical nature of the coupled process as described in the reports on current effect on interfacial reactions and synthesis of bulk materials by current-assisted power metallurgy (ref. 9-14).

Further still, I see no time dependent potential in this relationship. The authors introduce the parameter j as: "...the current density.... In the DC mode, there is no couplable j , so the atom fluxes for Al and Fe during DC-aluminizing are reasonably

determined by the CPG. Whereas in the PDC mode, j is no longer zero. Although an exact magnitude for j is still missing, which may be estimated by solving the Maxwell equations in anisotropic solids[15], we may still gain a qualitative evaluation about the influences of j on the atom fluxes of Al and Fe." Again, ref [15] is cited and no rationale is given for why this parameter j is a function of dE/dt (AC vs DC). Instead, vague terms are used to explain this parameter ("no couplable j ").

Actually, the parameter j in the second item, which is the branch component of the eddy current that flows perpendicular to the surface, is directly related to the time dependent potential dE/dt . It is known that the eddy current is produced by the changing magnetic field dB/dt . Meanwhile, the changing magnetic field dB/dt is produced by the applied time dependent current (PDC or AC) dj_{ap}/dt , i.e. the dE/dt . Hence, the parameter j is a function of dE/dt .

Further explanation has been added now in the manuscript as follows: In a simplified manner, there is a proportional relation between j and the time-dependence of the applied current dj_{ap}/dt . According to Ampère's law, the applied time-dependent current dj_{ap}/dt produces a time-dependent magnetic field dB/dt which is numerically proportional to dj_{ap}/dt . Meanwhile, the changing magnetic field generates an eddy current j_{ed} which is numerically proportional to dB/dt , according to Faraday's law. Since the current j is one branch of the eddy current j_{ed} that flows perpendicularly to the metal surface. A simplified relation can be obtained that j is proportional to the time dependence of the applied current dj_{ap}/dt .

The term "no couplable j " has been modified in the manuscript now. "there is no couplable j " is now replaced "there is no current that flows in parallel with the CPG, i.e. $j = 0$ ".

We have carefully checked the manuscript. The manuscript has been resubmitted to your journal. We look forward to your positive response.

Sincerely,

Mingli Shen

Reviewers' Comments:

Reviewer #2 (Remarks to the Author):

The authors have sufficiently answered the questions raised by the review process and the manuscript should be accepted by the journal.

Reviewer #3 (Remarks to the Author):

I read the rebuttal by the authors, yet I am not recommend the revised paper for publication. This is because there is very little science in the paper. Electromigration is a time-dependent event. What has happened may be due to Joule heating under an extremely large current. It is a fast surface modification technique for alloys.